# Understanding the effect of human capital and decent work for migrants' integration using PLS-SEM

Mirza Marvel Cequea[1], Jessika Milagros Vásquez Neyra[2],
Valentina Gomes Haensel Schmitt[3], *

1 Escuela de Ciencias Empresariales, ECIEM, Universidad Católica del Norte, Coquimbo, Chile,
2 CENTRUM Católica Graduate Business School, Pontificia Universidad Católica del Perú, Lima, Perú,
3 Escuela de Negocios y Economía, Pontificia Universidad Católica de Valparaíso, Valparaíso, Chile

* valentina.schmitt@pucv.cl

## Abstract

This study investigates the impact of human capital on the integration of Venezuelan migrants in Peru, considering decent work as a crucial mediator. The research involved a sample of 1,193 Venezuelan adults residing in Lima and applied Partial Least Squares Structural Equation Modeling to examine the relationships among human capital, decent work, and integration. Findings reveal that human capital significantly enhances migrant integration, facilitating their participation in the labor market and contribution to the local economy. Decent work emerged as a key factor in promoting integration by providing economic stability and supporting social inclusion. However, Peru's high level of labor informality restricts migrants' access to formal and dignified employment, limiting the potential impact of decent work on integration. These results underscore the importance of policies aimed at labor formalization and the recognition of migrant skills, which could maximize migrants' contributions and foster social cohesion. This study provides insights relevant for policymakers in Latin America, particularly in contexts with high labor informality, to develop effective strategies for the integration of migrant populations.

## 1. Introduction

International migration has grown significantly in recent decades, affecting both origin and destination countries [1]. In Latin America, Venezuelan migration constitutes the region's largest forced displacement [2], often described as an exodus [3,4]. This phenomenon presents important challenges for the institutions and governments of host countries, which must respond to the needs of migrants and facilitate their integration [5,6].

The integration of migrants can bring economic and social benefits to host societies by fostering social cohesion and economic and cultural development [7]. Key

**Data availability statement:** https://dataverse.harvard.edu/dataset.xhtml?persistentId=doi:10.7910/DVN/UHGMYG.

**Funding:** The author(s) received no specific funding for this work.

**Competing interests:** The authors have declared that no competing interests exist.

factors in this process include human capital and decent work. Human capital, understood as the skills and knowledge of migrants, facilitates their labor insertion and social contribution [8]. In turn, decent work, defined by the International Labour Organization (ILO) as employment with fair conditions and social protection [9], is essential to achieving sustainable and dignified integration [10].

One of the main challenges is to provide employment opportunities in settings that often have limited resources to meet these demands [11].

Peru has become the second largest recipient of Venezuelan migrants in Latin America [12]. This migratory flow has created challenges for a country with limited experience in managing international migration, underscoring the importance of legal frameworks that promote integration and protect migrants' rights [1,5,6,13]. The Peruvian economy has also been impacted, with an estimated GDP increase of 0.08% per year between 2018 and 2019 due to Venezuelan immigration [14].

The arrival of migrants has increased pressure on public services and infrastructure [15] and has raised participation in the informal sector, within a market already characterized by high informality [16,17]. This has spurred efforts to reduce migrants' vulnerabilities, leverage their human capital, and foster decent working conditions [18].

The objective of this study is to analyze the impact of human capital on the integration of Venezuelan migrants in Peru, with decent work as a key mediator in this relationship. This research is especially relevant in the Peruvian context, where the integration of Venezuelan migrants impacts social cohesion and the economy. Understanding the role of human capital and decent work in integration can guide policies that benefit both migrants and host communities, promoting a productive and cohesive environment.

The study addresses a gap in the migration literature by focusing on integration in contexts of high informality, such as Peru, and offers practical recommendations to improve working conditions and foster social cohesion. This research aims to establish a foundation for inclusive policies applicable in other Latin American contexts facing similar challenges.

## 2. Literature review and hypotheses

### 2.1. Human Capital and Integration

Human capital is understood as the set of productive skills and knowledge of a population, with educational level being a key indicator of this capital [19–21]. Historically, migrants were perceived as a burden on host countries due to their low educational levels, which limited their productivity [21]. In post-industrial economies, highly skilled migrants began to be valued, while those with low skills faced more obstacles [22].

Currently, the positive impact of highly skilled migrants on the economy of host countries is recognized, with their contributions in the business environment and innovation being particularly noted [23]. As countries recognize the benefits of talent migration [24,25], brain gain becomes an opportunity to boost productivity and economic development without incurring the costs of training [26–30].

In the case of Venezuelan migrants in Peru, many arrive with high academic qualifications but face barriers such as the lack of recognition of their credentials, which limits their labor integration. This phenomenon, known as brain waste, occurs when highly qualified migrants accept unskilled jobs or are unable to leverage their training in the labor market [4,28,31–36]. This represents a loss of opportunities not only for the migrants but also for the host country, which misses out on potential improvements in productivity and innovation.

Despite their high levels of human capital, many migrants face barriers such as cultural prejudice and racism, which hinder their access to adequate employment [37,38]. In this context, integration policies play a crucial role, and host countries can benefit from promoting the inclusion of these talents in their economy [20,39–42]. Countries adopt integration models that blend assimilation, segregation, or multiculturalism based on labor needs [20,43].

The situation of Venezuelan migrants in Peru illustrates the importance of effective integration. Peru, as the second-largest recipient of Venezuelan migrants in Latin America, faces integration challenges in its highly informal labor market, where approximately 70% of workers are in informal jobs [17]. In this regard, coordination among actors such as the State, NGOs, and the private sector is essential to facilitate the inclusion and utilization of migrant human capital [44–46]. Based on the aforementioned background, the following hypothesis is proposed:

> H1: The human capital of Venezuelan migrants has a positive and significant impact on their integration into the host society, as higher levels of skills and knowledge facilitate their entry into the labor market and participation in social life.

## 2.2. Human capital and decent work

Decent work is defined as productive employment that provides fair remuneration, security, social protection, and equal treatment [47–51]. This concept is a fundamental goal of the United Nations 2030 Agenda [51] and is particularly relevant for migrants, displaced persons, and refugees, who need their labor rights to be respected in host countries [52]. However, many migrants are forced to enter informal labor markets due to a lack of documentation, which limits the recognition of their skills and places them in vulnerable positions with low wages [51,53–57].

Research shows that low-income migrants often face reductions in their labor rights and working conditions due to their socio-legal precariousness and the temporary nature of their jobs, which can lead to situations bordering on modern slavery [49,52,54,55]. These conditions may include harassment, forced labor, and unfair remuneration.

To fully harness the human capital of migrants, it is essential that host countries overcome barriers such as the recognition of skills and professional qualifications. This is crucial to reduce vulnerability and facilitate the integration of migrants into the formal economy, promoting brain gain and avoiding brain waste [16,27,58].

In summary, migrants face significant barriers to accessing decent work in host countries. Therefore, the following hypothesis is proposed:

> H2: The human capital of Venezuelan migrants positively and significantly influences the achievement of decent jobs, as a high level of skills and education is expected to increase opportunities for access to adequate and secure employment.

## 2.3. Decent work and integration

Decent work is essential for the integration of migrants, both economically and socially. Integration involves not only labor market insertion but also access to housing, education, healthcare services, and the exercise of civic rights [52,57]. According to Khan & Sandhu [59], promoting decent work practices and understanding migrant cultures can improve migration policies and facilitate their integration.

Social integration represents a sense of belonging and cohesion, achieved through cultural adaptation and the creation of support networks [60,61]. Economic integration, on the other hand, is facilitated by stable employment that allows for

 

the coverage of basic needs and provides financial stability [16]. Fair remuneration, as a central element of decent work, contributes to improving quality of life and supports an effective integration process [10,56].

In the case of Peru, many Venezuelan migrants are in informal jobs where their educational level does not predict the type of employment they obtain, leading them to work in positions for which they are overqualified [16,62]. This highlights the need to establish decent work conditions that foster their integration. In this context, decent work can be a key factor in generating a sense of belonging among migrants. Based on the above, the following hypothesis is proposed:

> *H3: Decent work has a positive and significant impact on the integration of Venezuelan migrants*, providing them with economic stability, security, and opportunities to establish social networks, which collectively strengthen their sense of belonging.

### 2.4. Mediation role of decent work

Education and employment are key indicators of integration [7,47], and migratory flows can strengthen the workforce in host societies [63]. However, integration is only viable with access to decent work [47,52]. Venezuelan migration in South America illustrates how working conditions influence social and economic inclusion, as migrant unemployment surpasses that of the local population [64].

Obtaining formal employment, facilitated by state support and business flexibility, is essential for integration. Migrants without work permits often find themselves in informal, precarious jobs without access to social benefits [65,66]. Implementing formal integration policies benefits both society and the economy, as legally employed migrants contribute through taxes and consumption [44].

According to the ILO [48], decent work includes social protection, labor rights, job security, and fair remuneration, allowing migrants to convert their human capital into productive work and facilitating their integration [47,50,51]. Securing decent employment provides economic stability, improves quality of life, and fosters the learning of social and cultural values, promoting social and cultural integration [47,60,61].

In summary, decent work acts as a mediator between human capital and integration, facilitating economic and social integration. Without dignified employment, even highly qualified migrants can be marginalized, limiting their capacity for integration. Therefore, the following hypothesis is proposed:

> *H4: Decent work positively and significantly mediates the relationship between human capital and the integration of Venezuelan migrants*, offering a context in which migrant's skills can be fully leveraged to support their integration process. This mediation underscores the importance of decent work as a pathway for skilled migrants to achieve sustainable inclusion in the host society.

Fig 1 illustrates the proposed research hypotheses, where the human capital of Venezuelan migrants promotes integration and access to decent work in the host country; and decent work plays a mediating role between human capital and integration in their new environment.

## 3. Materials and methods

### 3.1. Sampling procedure

This study was conducted using a quantitative approach, based on data obtained from a publicly available database in Harvard Dataverse, corresponding to a study by Cequea et al. [67]. Data collection was carried out through a survey conducted in Lima, the capital of Peru, in 2022, yielding a total of 1,193 valid responses. The sample consisted of Venezuelan adults residing in the urban area of Lima, including both men and women, professionals and non-professionals, who visited an NGO that provides support to migrants and refugees, primarily Venezuelans [68].

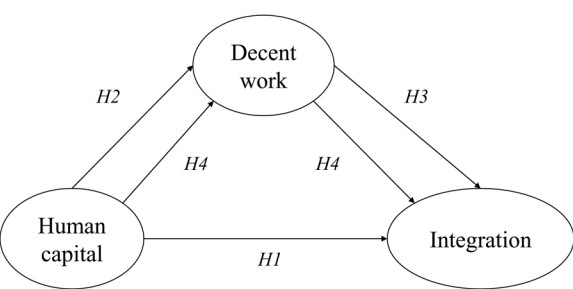

**Fig 1. Research model.**

### 3.2. Data used

The data used in this study was extracted from the Harvard Dataverse database, 'Data for the project: Human Capital and Decent Work: Venezuelan Migrants in Peru as Productive Subjects in the Labour Market: The Case of Metropolitan Lima' [68].

This study was based on secondary data and therefore does not raise ethical issues regarding the participation of human subjects. The data used originated from the study by Cequea et al. [67], which was reviewed and approved by the ethics committee of the University of Lima, Peru [68]. According to Cequea et al. [68], the data set was anonymized during collection, with all personally identifiable information removed, ensuring confidentiality and protection of the participants' data. Before completing the questionnaire, participants gave written consent, as required by ethical protocols. The study received ethical approval from the ethics committee of the University of Lima -Dictamen N. 027-CEI-IDIC-IDIC-ULIMA-2022- [68]. By using secondary data from a publicly available and ethically reviewed source, this study adheres to international ethical standards for research, including the Declaration of Helsinki. These measures ensure the safeguarding of sensitive migration data and address potential ethical issues related to participant confidentiality.

### 3.3. Measurements

This study considered three variables: human capital, decent work and integration. The variables were measured using a Likert-type scale from 1 (strongly disagree) to 5 (strongly agree) [67]. Four human capital items, eight decent work items and seven integration items are used in this study. For human capital, respondents assessed their work experience, education and skills; for decent work, respondents were asked to determine their productive work, job security, respect for human dignity and fair conditions; and for integration, respondents assessed their resilience, support networks and financial solvency to make commitments.

### 3.4. Data analysis

The data analysis process involved several steps to ensure data quality and robust results. After cleaning the data by removing outliers and imputing missing values, the study applied Partial Least Squares Structural Equation Modeling (PLS-SEM). This technique is particularly suited for predictive research involving complex models with latent variables. PLS-SEM was selected for its flexibility in handling non-normal data distributions—a characteristic often observed in datasets from migrant populations—and its ability to analyze multiple dependent variables simultaneously. These features align with the study's objective of exploring the interconnected effects of human capital, decent work, and integration [69,70].

To enhance the predictive validity of the findings, the study employed PLSpredict, a procedure used to evaluate the model's predictive relevance by generating predictions for unseen data, focusing on $Q^2$ and Standardized

Root Mean Square Residual (SRMR) as key indicators. Q² measures the model's predictive relevance, while SRMR evaluates its fit by examining the residuals between observed and predicted correlations. In this study, SRMR values remained within acceptable thresholds, confirming the model's ability to accurately represent the data structure. This analysis validated both the relationships between variables and the model's predictive accuracy for new data [71,72].

The study adhered to established PLS-SEM evaluation guidelines [70,71,73]. Both the measurement model – which evaluates construct reliability and validity – and the structural model – which examines relationships between variables – were rigorously tested. This comprehensive approach enhances the policy relevance and practical applicability of the findings for migrant integration.

### 3.5. Addressing common method bias (CMB)

To address potential Common Method Bias (CMB) and ensure research validity, the study implemented preventive measures during its design phase, following the recommendations of Schwarz et al. [74]. Additionally, statistical tests such as the lateral collinearity test confirmed that CMB did not significantly influence the results [75]. The analysis yielded a variance inflation factor (VIF) of 1.268, well below the threshold of 3.3 [76], further validating the robustness of the model.

### 3.6. Ethical statement

This study was based on secondary data and therefore does not raise ethical issues regarding the participation of human subjects. The data used originated from the study by Cequea et al. [67], which was reviewed and approved by the ethics committee of the University of Lima, Peru [68]. According to Cequea et al. [68], the data set was anonymized during collection, with all personally identifiable information removed, ensuring confidentiality and protection of the participants' data. Before completing the questionnaire, participants gave written consent, as required by ethical protocols. The study received ethical approval from the ethics committee of the University of Lima -Dictamen N. 027-CEI-IDIC-IDIC-ULIMA-2022- [68]. By using secondary data from a publicly available and ethically reviewed source, this study adheres to international ethical standards for research, including the Declaration of Helsinki. These measures ensure the safeguarding of sensitive migration data and address potential ethical issues related to participant confidentiality.

## 4. Results

### 4.1. Descriptive statistics

Fig 2 provides a detailed overview of the sample's demographic characteristics. Women represent the majority of the sample (62%), while men comprise 38%. Regarding age distribution, the largest group falls within the 31–40 age range (37% for both genders), followed by participants aged 21–30 (28% of women and 34% of men). A smaller proportion of participants are above 50, with only 6% of men and 10% of women in the 51–60 age group, and even fewer aged 61 or older.

Marital status indicates that single individuals dominate the sample, accounting for 70% of women and 66% of men. Married participants represent 18% of women and 22% of men, while divorced and widowed individuals make up a small minority.

In terms of education, the sample reflects high human capital. Among women, 41% hold university degrees, compared to 34% of men. Postgraduate qualifications are less common, with 6% of women and 3% of men attaining this level. A notable proportion of men have secondary (40%) or technical education (22%), compared to 32% and 21% of women, respectively.

Employment patterns reveal a significant prevalence of independent work, particularly among women (46%) compared to men (36%). Meanwhile, 37% of women and 43% of men are employed without formal contracts, underscoring the informal nature of employment within the sample. A smaller subset of participants is employed under contractual agreements, with slightly more men (22%) than women (18%).

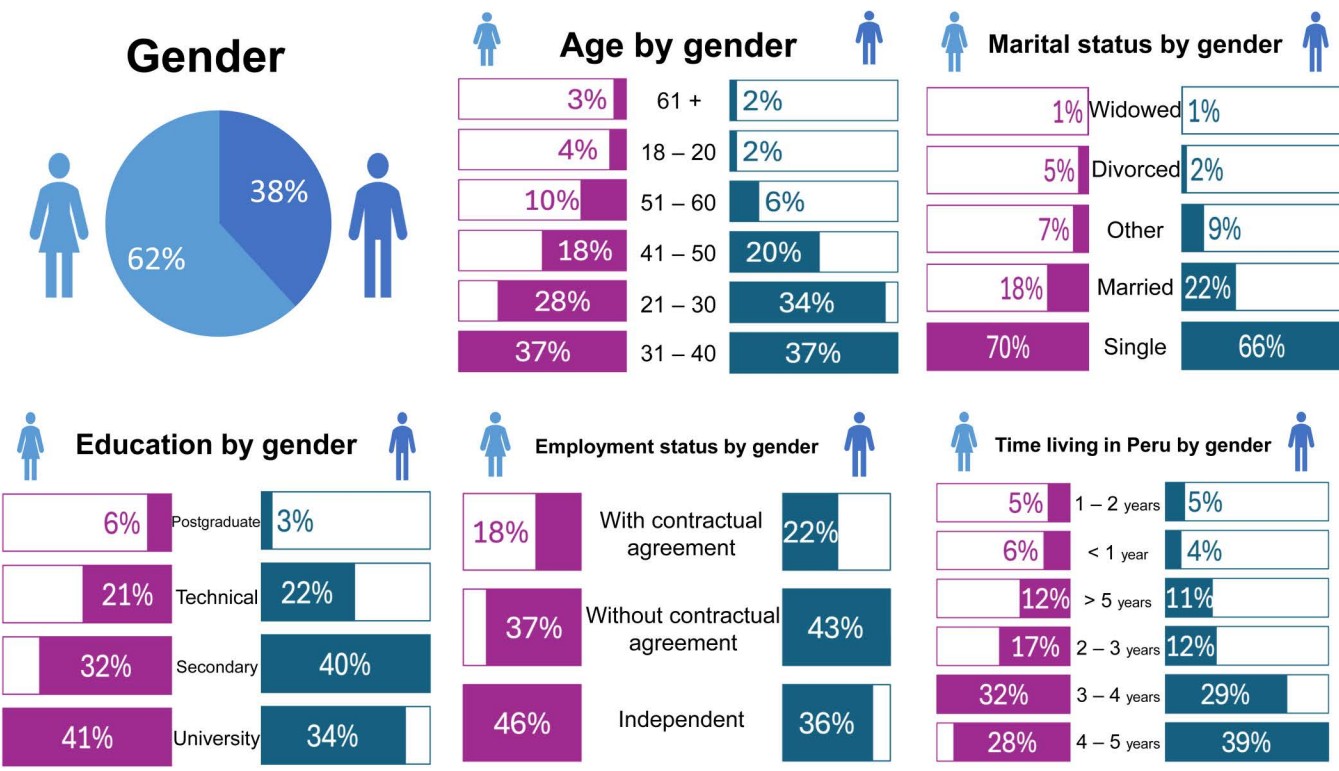

**Fig 2. Characteristics of participants.**

Residency duration in Peru varies across the sample. Most participants have lived in the country for three to five years, with 32% of women and 29% of men reporting 3–4 years of residence, and 28% of women and 39% of men indicating 4–5 years. These patterns suggest a mix of recently arrived individuals and those who are more established, which impacts labor market integration and stability.

Additional data highlight that 65% of respondents have 1–10 years of professional experience, indicating a relatively experienced workforce. Work intensity is notable, with 59% of participants working six days per week and 38% reporting shifts of 10–12 hours daily. This reflects the demanding nature of their work environments and economic necessity.

Lastly, 51% of respondents report practicing their profession in Peru, demonstrating a level of professional integration. However, this also implies that nearly half face barriers in exercising their qualifications, likely due to issues related to credential recognition or occupational mismatch.

Additionally, 51% of respondents are able to practice their profession in Peru, highlighting a degree of professional integration. However, this also suggests that nearly half face barriers to exercising their qualifications in the local job market, potentially due to credential recognition or occupational matching challenges.

## 4.2. Evaluation of the measurement model

A reflective measurement model was applied, following the recommended guidelines for evaluating reliability and validity [70]. Table 1 presents detailed results, including loadings, Cronbach's alpha (α), composite reliability (CR), average variance extracted (AVE), and discriminant validity through HTMT values.

**Table 1. Reliability and Validity Measures for Measurement Model Constructs.**

| Construct | Indicator | Loading | a | CR | AVE |
|---|---|---|---|---|---|
| Human capital | HC1 | 0.920 | 0.930 | 0.948 | 0.786 |
| | HC2 | 0.922 | | | |
| | HC3 | 0.895 | | | |
| | HC4 | 0.924 | | | |
| | HC5 | 0.759 | | | |
| Decent Work | DW1 | 0.898 | 0.944 | 0.955 | 0.781 |
| | DW2 | 0.888 | | | |
| | DW3 | 0.851 | | | |
| | DW4 | 0.869 | | | |
| | DW5 | 0.889 | | | |
| | DW6 | 0.907 | | | |
| Integration | INT1 | 0.890 | 0.962 | 0.969 | 0.817 |
| | INT2 | 0.848 | | | |
| | INT3 | 0.894 | | | |
| | INT4 | 0.905 | | | |
| | INT5 | 0.918 | | | |
| | INT6 | 0.939 | | | |
| | INT7 | 0.929 | | | |

Note(s): α: Cronbach's Alpha, CR: Composite Reliability, AVE: Average Variance Extracted.

Source: own elaboration.

**4.2.1. Reliability assessment.** Item reliability: All items demonstrated sufficient individual reliability, with outer loadings above 0.818, exceeding the recommended threshold of 0.708 [70]. This indicates that each item has a high correlation with its associated construct.

Internal consistency: The Cronbach's alpha values for each construct were all above the 0.7 threshold, demonstrating satisfactory internal consistency. Specifically, human capital achieved an α of 0.930, decent work 0.944, and integration 0.962, confirming the internal coherence of items within each construct [70].

Composite Reliability: CR values also surpassed the recommended 0.7 threshold, with human capital at 0.948, decent work at 0.955, and integration at 0.969. This further supports the internal consistency of the constructs, indicating that the items provide consistent measurements of their respective latent constructs [70].

**4.2.2. Validity assessment.** Convergent validity: Assessed through the Average Variance Extracted (AVE), all constructs achieved AVE values above 0.5. human capital showed an AVE of 0.786, decent work 0.781, and integration 0.817, meeting the criterion set by Fornell and Larcker [77]. This indicates that more than 50% of the variance in each construct's indicators is explained by the construct itself, confirming good convergent validity.

Discriminant Validity: The Heterotrait-Monotrait (HTMT) ratio was used to assess discriminant validity. All HTMT values were below 0.85, as recommended by Henseler et al. [78], ensuring that each construct is empirically distinct from the others.

These comprehensive reliability and validity assessments confirm that the measurement model is robust, meeting all recommended thresholds in the literature [70,77,78], and thereby supports the structural model's capacity to test the proposed hypotheses.

## 4.3. Evaluation of structural model

The structural model evaluation followed four steps [70]: (1) addressing collinearity using the Variance Inflation Factor (VIF), (2) assessing structural relationships significance and relevance via path coefficients (β) and p-values, (3) evaluating explanatory power with R², adjusted R², and the Standardized Root Mean Square Residual (SRMR), and (4) examining predictive power through the Q². The findings are summarized in Table 2 and illustrated in Fig 3.

Fig 3 provides an overview of the path coefficients, displaying their direction, magnitude, significance levels, and R² values, offering a comprehensive picture of the relationships within the structural model. The VIF values for all constructs, which ranged from 1.000 to 1.268, indicate an absence of multicollinearity issues, ensuring the robustness of the model

**Table 2. Structural model.**

| Coefficient | t-value | p value | | | |
|---|---|---|---|---|---|
| 0.788 | 32.327 | 0.000 | | | |
| **Coefficient** | **t-value** | **p value** | **Results** | **VAF** | |
| 0.746 | 21.781 | 0.000 | Supported | 95,54% | |
| 0.459 | 11.838 | 0.000 | Supported | | |
| 0.092 | 1.732 | 0.019 | Supported | | |
| **Coefficient** | **5% PBCI** | **95% PBCI** | **Results** | **VAF** | **v** |
| 0.042 | 0.009 | 0.077 | Supported | 4.46% | 0.002 |

**Note(s):** Bootstrapping based on n = 10,000 subsamples. Mediating effects are assessed by applying a one-tailed test. HC: Human capital, INT: Integration, DW: Decent work, PBCI: Percentile bootstrap confidence interval, SRMR: Standardized root mean square residual, VAF: Variance accounted for, **v**: Effect size for mediation analysis.

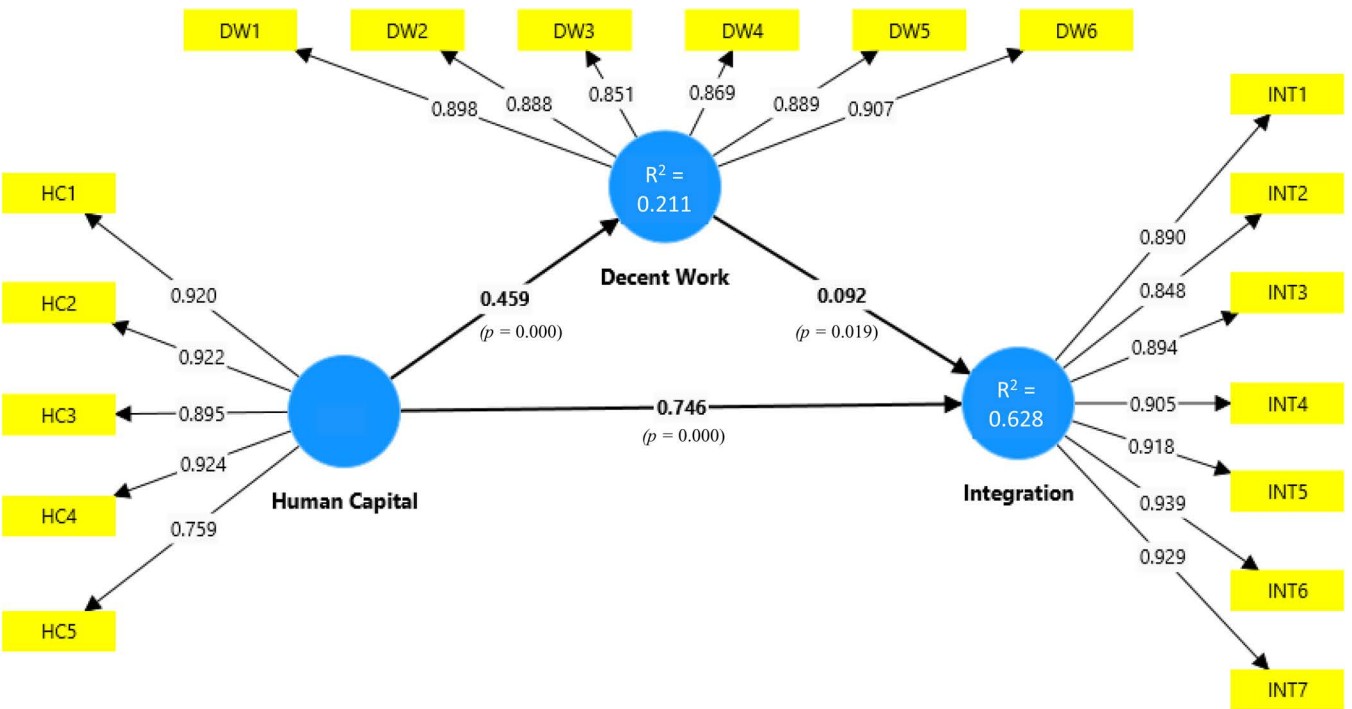

**Fig 3. Partial least square SEM model.**

[71]. Bootstrapping, a resampling method that generates multiple random subsamples from the dataset to enhance the reliability of statistical estimates, was employed with 10,000 samples that was used to calculate t-statistics and confidence intervals, enhancing the reliability of the significance testing for each path [71].

Hypothesis H1 was supported. The path coefficient for human capital on integration was $\beta = 0.746$ ($p < 0.05$), indicating a strong, significant, and positive impact. A high $\beta$ value indicates that increased levels of human capital (e.g., education, skills) strongly contribute to integration within the host society. This finding underscores the relevance of policies promoting human capital development, like educational and vocational training, to ease the migrant integration process.

Hypothesis H2 was supported. Human capital positively impacts decent work with a coefficient of $\beta = 0.459$ ($p < 0.05$). This $\beta$ value suggests that human capital enhances the likelihood of securing decent work, which includes aspects such as fair remuneration, job stability, and safe working conditions. This finding is particularly important in contexts with high informal employment, like Peru, where decent work remains a pathway to integration.

Hypothesis H3 was supported. Decent work has a significant but smaller effect on integration, with $\beta = 0.092$ ($p < 0.05$). Although the coefficient is lower than that of human capital, it reveals that decent work facilitates integration by providing economic stability and social inclusion. Decent work not only contributes to financial stability but also enhances migrants' opportunities for social participation, which is essential for long-term integration.

Hypothesis H4 was supported. The mediation analysis followed the approach of Nitzl et al. [79], examining the total, direct, and indirect effects of human capital on integration. Using a bootstrapping procedure with 10,000 samples and percentile confidence intervals, the analysis confirmed that decent work partially mediates the relationship between human capital and integration. Although the mediation effect is low, it is significant, suggesting that decent work contributes incrementally to integration outcomes. The continued significance of the direct effect of human capital on integration implies partial mediation, highlighting that, while human capital has a strong direct impact, decent work enhances this effect by providing additional support for the integration process.

To further interpret the mediation, the variance accounted for (VAF) index was calculated based on Alwin and Hauser [80] and Henseler [78] (see Table 2, Fig 4), contrasting the indirect and direct effects in relation to the total impact. The VAF index, which quantifies the proportion of an effect that is mediated, indicates that 4% of the effect of human capital on integration is mediated by decent work, with the remaining 96% representing a direct impact. This low mediation percentage suggests that, although decent work plays a role, human capital independently drives most of the integration effect. However, even a partial mediation effect is practically relevant, as it underscores the importance of decent work in enhancing social and economic inclusion for migrants.

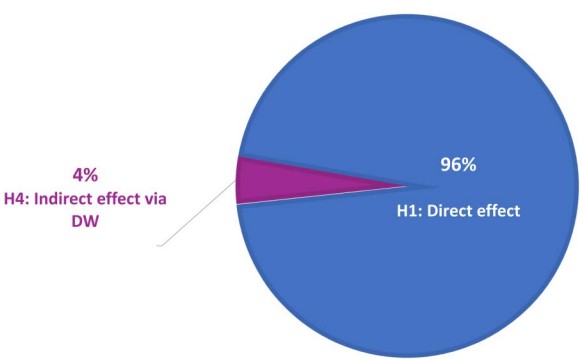

**Fig 4. Partitioning the total effect of Human Capital on Integration using VAF.**

The effect size, calculated as recommended by Lachowicz et al. [81] and interpreted following Gaskin et al. [82], yielded a v value of 0.002, indicating a small effect size. Despite this small size, the mediation effect of decent work is meaningful, as it reflects the necessity of fair and stable employment in bolstering the integration process. Access to decent work not only augments the economic stability of migrants but also fosters social cohesion, reinforcing the idea that successful integration requires a combination of skill development and employment opportunities.

The structural model showed moderate explanatory power, with an R² value of 0.628 for integration. This result suggests that human capital and decent work collectively account for 62.8% of the variance in integration, which aligns with a substantial portion of the variability explained by the model. Additionally, the SRMR value of 0.037, which is well below the commonly accepted threshold of 0.08, further supports a good model fit. This low SRMR value indicates that the observed data aligns closely with the model's predicted values, strengthening the argument for the model's suitability in representing the relationships within the dataset.

The model's predictive power was assessed using the PLSpredict [83], with integration as the target construct. A k-fold cross-validation (k = 10), a method that splits the dataset into k subsets to validate the model across different samples, ensuring a minimum of 30 cases in each holdout sample, produced a positive Q² value of 0.619, indicating substantial predictive relevance. This Q² result, alongside the SRMR value, confirms both a strong predictive capacity and a good model fit. Together, these metrics validate the model's robustness and suitability for analyzing migrant integration through human capital and decent work, suggesting it can be reliably applied to similar populations beyond the sample data.

## 5. Discussion

The proposed structural model follows a reflective approach, and the formulated hypotheses [H1, H2, H3, and H4] have been supported by the results obtained. The main findings are as follows:

### 5.1. Direct impact of human capital on integration [H1]

The results confirm hypothesis H1, demonstrating a positive and significant relationship between human capital and migrant integration. This finding is consistent with previous studies [7,25,29,37,43], which emphasize that migrants' skills and knowledge facilitate their integration into the host economy and society. In the Peruvian context, where labor informality presents challenges, human capital stands out as a key factor that could enhance integration if competency recognition policies are in place. This finding suggests that the development of training programs and competency validation could be fundamental in maximizing the positive impact of human capital on integration.

Additionally, the structural model highlights specific critical factors for migrants, such as concerns for professional growth, skills retention, and recognition of prior experience. These findings align with the concept of brain waste, where the underutilization of migrant skills hinders their integration and potential contribution [4,28,31–36].

### 5.2. The role of human capital in accessing decent work [H2]

The results confirm hypothesis H2, showing that migrants' human capital is a determining factor in obtaining decent jobs. Migrants with higher education and experience levels are more likely to access formal and stable jobs [19]. However, the results also reveal that many migrants in Peru, despite being highly qualified, are forced to work in informal jobs, limiting their development and reflecting a brain waste phenomenon. This finding underscores the need for policy interventions that promote labor formalization and remove barriers to decent employment for skilled migrants, thus allowing them to realize their potential in the local economy.

Decent work enables migrants to enhance their skills and access better job opportunities, thereby improving their socioeconomic status. However, the low contribution observed in this study could reflect the precarious conditions experienced by many migrants in Peru, which limit their professional development opportunities [16,38,45].

### 5.3. Contribution of decent work to migrant integration [H3]

The results confirm hypothesis H3, indicating that decent work facilitates migrant integration by providing economic stability, access to basic needs, and the opportunity to contribute to the tax system [24,25,29,44]. Employment in fair conditions not only improves economic security [52] but also promotes social cohesion by reducing negative stereotypes and fostering inclusion [23,37]. This finding suggests that ensuring decent working conditions for migrants is crucial to facilitating their integration and strengthening social cohesion in the host country.

Moreover, decent work fosters the development of new skills and professional opportunities, allowing migrants to significantly contribute to the local economy and bolster public services through social security contributions [25,44]. Social and community participation, such as collaboration with local organizations, is a key factor for integration, suggesting that decent work can indirectly support integration by strengthening social networks [60,61].

### 5.4. Mediating role of decent work between human capital and integration [H4]

The results support hypothesis H4, showing that decent work acts as a mediator in the relationship between human capital and integration, though with a limited effect due to the high labor informality in Peru. This implies that, although decent work enhances the impact of human capital on integration, its influence is restricted in contexts where informal employment prevails [51,53–55].

Informality in Peru limits the potential of decent work as a mediator in migrant integration, as it prevents access to social benefits and job stability, which are key elements for economic and social inclusion. This finding highlights the importance of policies promoting formal employment and dignified working conditions, enabling migrants to maximize their skills and contribute fully to the host society.

For host countries, facilitating access to decent work for skilled migrants represents a brain gain opportunity, where the host country benefits from human capital without bearing the costs of education [28,30]. However, barriers to integration and the prevalence of precarious working conditions can lead to brain waste, limiting both the economic contributions and social integration of skilled migrants [27]. Policies aimed at bridging these gaps are essential to realize the dual benefits of human capital development and migrant integration.

According to the International Labor Organization [9], decent work implies access to productive opportunities in conditions of freedom, equity, security and human dignity, essential aspects for the social and economic integration of migrants in host societies, since it not only guarantees stable income and decent working conditions, but also promotes their social recognition, civic participation and general well-being. Likewise, access to employment with fair conditions allows migrants to develop economic autonomy, facilitating their stability and access to housing, education and health services. Additionally, it encourages the use of skills and knowledge acquired in the country of origin, avoiding overqualification and waste of talent, which reinforces their self-esteem and sense of belonging [57]. Finally, the workplace is a space where migrants can establish contact networks, learn cultural norms and develop interpersonal relationships that favor their social integration [7]. In this sense, inclusive labor policies strengthen social cohesion and economic stability, benefiting both migrants and society as a whole, since when host countries guarantee fair labor conditions, they not only protect migrants' rights, but also take advantage of their human capital and promote more cohesive and equitable societies [10].

Another aspect that could be affecting the low mediation of decent work are the barriers faced by migrants in Peru to regularize their qualifications and validate their knowledge and experience, finding themselves limited to underemployment and not being able to access well-paid jobs commensurate with their skills, which reduces their quality of life [16,62].

### 5.5. Challenges for implementing labor formalization policies in the Peruvian context

Despite the need for policies that promote labor formalization to facilitate migrant integration, their implementation in Peru faces significant structural challenges. High labor informality reflects not only a lack of access to formal jobs but also

issues such as bureaucracy and the lack of economic incentives for employers and workers. For instance, the high costs associated with formally registering employees, along with the administrative burden of complying with complex labor regulations, discourages many companies, especially small and medium-sized ones, from formalizing their workforce. This is compounded by limited technological infrastructure in some areas, which complicates access to digitalized formalization processes.

These barriers disproportionately affect migrants, who often lack the social networks or resources to navigate these processes effectively. Moreover, the prevalence of informal employment reduces opportunities for skilled migrants to access jobs that align with their qualifications, limiting their potential contribution to the formal economy.

Additionally, the lack of an efficient system for recognizing foreign degrees and professional experience remains a significant barrier to the integration of skilled migrants. Without a streamlined accreditation process, many highly qualified migrants are unable to practice their professions, fostering brain waste and limiting their economic contribution. This challenge is particularly acute in sectors such as health and education, where accreditation is essential for employment.

### 5.6. Strategies to overcome these challenges

To address these challenges, several strategies could be implemented. First, simplifying administrative procedures and reducing the costs associated with formal employee registration could encourage more companies to formalize their workforce. Examples from countries such as Colombia and Chile, which have implemented tax incentives and simplified digital platforms for formalization, could serve as models for Peru [42]. Or strategies for rapid regularization of their stay in the context of arrival, as in the case of Ecuador, so that they can access formal employment, the public health system, rental housing and banking, among others [39]. This might include creating tax incentive programs for employers who formally hire skilled migrants, as well as subsidies to cover part of the formalization costs in strategic sectors.

Second, creating a flexible and accessible accreditation system to recognize foreign degrees would be essential. This could be achieved through cooperation agreements with foreign universities and educational institutions, as well as through the development of practical assessments that allow migrants to directly demonstrate their competencies.

Finally, establishing centralized digital platforms where migrants can access information on formalization opportunities and legal resources would simplify the process, especially in areas where access to technological infrastructure is limited. Collaboration between the government, NGOs, and the private sector could facilitate the dissemination and access to these resources, thus promoting a more effective and productive integration. Such platforms could also serve as hubs for professional networking and mentorship opportunities, which are critical for skilled migrants navigating new labor markets.

## 6. Conclusions

This study highlights the pivotal role of human capital in the integration of Venezuelan migrants in Peru, emphasizing decent work as a critical mediating factor. This research demonstrates that human capital directly and positively impacts the integration of Venezuelan migrants in Peru, facilitating their labor market participation and economic contribution [H1]. This highlights the need for training and skills certification programs to enhance integration, reduce brain waste, and strengthen the host country's economy.

The study also links human capital to access to decent work [H2], though Peru's high labor informality restricts formal employment opportunities for migrants, underscoring the need for policies that promote labor formalization, respect labor rights, and fully utilize migrants' skills.

Decent work is also essential for the social and economic integration of migrants [H3], as it provides stability and allows them to meet basic needs, strengthening their sense of belonging. Policies ensuring dignified working conditions are essential for migrant integration and bolster social cohesion within host communities.

Decent work partially mediates the effect of human capital on integration [H4], yet its impact remains constrained by Peru's high rate of informal employment. Policies aimed at creating formal jobs could maximize migrant potential and economic contributions.

### 6.1. Practical implications

These findings underscore the importance of strengthening migrants' human capital through training programs adapted to the Peruvian labor market, addressing both technical and cultural skills.

The positive relationship between decent work and integration underscores the need for better working conditions, stability, fair wages, and access to social benefits. Authorities should enforce labor regulations that ensure an inclusive and fair work environment for migrants.

To counteract high labor informality, it is crucial to streamline the recognition of foreign degrees and make work permits more flexible, facilitating migrants' access to formal employment. This benefits not only the migrants but also addresses labor demands in key sectors and contributes to the economy by increasing tax revenues.

Awareness campaigns highlighting migrants' contributions are fundamental for enhancing public perception and reducing xenophobia, promoting a more inclusive society. Collaboration among the government, NGOs, and the private sector is essential to disseminate these messages and strengthen social cohesion.

Finally, both the State and the private sector can benefit from brain gain by integrating skilled migrants into the formal labor market, avoiding brain waste. This enables the capitalization of a skilled human resource without the need to invest in their training, thereby strengthening the national economy and productivity.

### 6.2. Limitations and future research directions

This study provides valuable insights into the social and economic integration of Venezuelan migrants in Lima, capturing relevant aspects that can inform public policies and integration strategies in similar contexts. While the sample focuses on Venezuelan migrants in one urban area, restricting generalizability to other contexts or migrant groups, its size is sufficient to draw meaningful conclusions for comparable urban settings in Latin America. The lack of gender parity among respondents, due to the sampling methodology at an NGO, reflects the demographic composition of the population seeking institutional support, adding practical relevance to the study's findings. However, this gender imbalance also highlights a potential limitation, as integration outcomes may vary significantly across different gender and age groups, which should be considered in future analyses.

On the other hand, a limitation of secondary research is that the data were collected for another purpose and may not fully meet the needs of the current research. The objectives and methodology used to collect the primary data may not be appropriate for the problem at hand, so the quality of the data source must be carefully reviewed [84,85]. For example, in the primary data collection [67], the data collection technique used did not guarantee parity between men and women, and the sampling was not random, so it is not possible to ensure representativeness. This could make it difficult to reach more robust conclusions if the objective of our study were aimed at making comparisons between groups of men and women; however, this was not the objective of the present study. On the other hand, to avoid assuming different meanings of the variables used for this secondary study, we reviewed the conceptualization and operationalization of these variables in the primary study [68] and adopted them for the secondary study.

While this study focused on the social and economic aspects of integration, the findings suggest opportunities for exploring other dimensions, such as cultural integration, across diverse contexts and with varied methodological approaches. Future studies could broaden the geographic scope and adopt a longitudinal design to assess how the roles of human capital and decent work evolve at different stages of the settlement process, and how factors like age and gender impact these roles over time. These extensions would not only provide a more comprehensive understanding of

migrant integration but also enhance awareness of the nuanced experiences of different demographic groups, complementing this study's findings and strengthening the knowledge base in the region.

## Author contributions

**Conceptualization:** Mirza Marvel Cequea, Jessika Milagros Vásquez Neyra, Valentina Gomes Haensel Schmitt.

**Data curation:** Mirza Marvel Cequea, Jessika Milagros Vásquez Neyra.

**Formal analysis:** Mirza Marvel Cequea, Jessika Milagros Vásquez Neyra, Valentina Gomes Haensel Schmitt.

**Investigation:** Mirza Marvel Cequea, Jessika Milagros Vásquez Neyra, Valentina Gomes Haensel Schmitt.

**Methodology:** Mirza Marvel Cequea, Jessika Milagros Vásquez Neyra.

**Project administration:** Mirza Marvel Cequea.

**Resources:** Mirza Marvel Cequea.

**Software:** Mirza Marvel Cequea, Jessika Milagros Vásquez Neyra.

**Supervision:** Mirza Marvel Cequea, Jessika Milagros Vásquez Neyra.

**Validation:** Mirza Marvel Cequea, Jessika Milagros Vásquez Neyra.

**Visualization:** Mirza Marvel Cequea, Jessika Milagros Vásquez Neyra, Valentina Gomes Haensel Schmitt.

**Writing – original draft:** Mirza Marvel Cequea, Jessika Milagros Vásquez Neyra, Valentina Gomes Haensel Schmitt.

**Writing – review & editing:** Mirza Marvel Cequea, Jessika Milagros Vásquez Neyra, Valentina Gomes Haensel Schmitt.

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
