## [Decision Letter · Decision Letter 0]

29 Jan 2025

PONE-D-24-51928Understanding the effect of human capital and decent work for migrants’ integration using PLS-SEMPLOS ONE

Dear Dr. Schmitt,

Thank you for submitting your manuscript to PLOS ONE. After careful consideration, we feel that it has merit but does not fully meet PLOS ONE’s publication criteria as it currently stands. Therefore, we invite you to submit a revised version of the manuscript that addresses the points raised during the review process.

We look forward to receiving your revised manuscript.

Kind regards,

Voxi Heinrich Amavilah, Ph.D.

Academic Editor

PLOS ONE

Journal Requirements:

2. Please note that your Data Availability Statement is currently missing the repository name. If your manuscript is accepted for publication, you will be asked to provide these details on a very short timeline. We therefore suggest that you provide this information now, though we will not hold up the peer review process if you are unable.

**Additional Editor Comments:**

Reviewer 2 has given you comments that would greatly improve your MS; please respond to them thoroughly. Like Reviewer 1, I would like to see some assessment of the wider implications of this Peruvian case study.

Reviewers' comments:

Reviewer's Responses to Questions

**Comments to the Author**

1. Is the manuscript technically sound, and do the data support the conclusions?

Reviewer #1: Yes

Reviewer #2: Yes

2. Has the statistical analysis been performed appropriately and rigorously? 

Reviewer #1: Yes

Reviewer #2: Yes

3. Have the authors made all data underlying the findings in their manuscript fully available?

Reviewer #1: Yes

Reviewer #2: Yes

4. Is the manuscript presented in an intelligible fashion and written in standard English?

Reviewer #1: Yes

Reviewer #2: Yes

5. Review Comments to the Author

Reviewer #1: The article is prepared at a fairly high level. The research topic is important and relevant.

It is worth adding an assessment of the representativeness of the sample of respondents.

It is worth adding the possibility of using the results in a business environment.

Reviewer #2: Review of Manuscript PONE-D-24-51928

Title: Understanding the Effect of Human Capital and Decent Work for Migrants' Integration Using PLS-SEM

General Comments:

This manuscript addresses a critical and timely issue: the integration of Venezuelan migrants in Peru, focusing on the roles of human capital and decent work. The study utilizes Partial Least Squares Structural Equation Modeling (PLS-SEM) to explore the relationships among these variables, providing insights relevant for policymakers and stakeholders in migration and labor markets. The manuscript is well-structured, and its findings are significant for understanding migrant integration in contexts characterized by labor informality. However, there are several areas where the manuscript could be improved to enhance its clarity, rigor, and impact.

Specific Comments:

1. Research Context and Contribution:

Strengths: The study's focus on Venezuelan migrants in Peru addresses a gap in the migration literature, particularly in the context of Latin America's high labor informality. The emphasis on decent work as a mediating factor is innovative and aligns well with the United Nations’ Sustainable Development Goals.

Suggestions for Improvement: While the manuscript discusses the relevance of the study, it could provide a more detailed comparison with similar studies in other regions to highlight its unique contribution further. For example, are there lessons from other countries facing similar migration challenges that could enrich the discussion?

2. Methodology:

Strengths: The use of PLS-SEM is appropriate for the study's objectives and the complexity of the model. The detailed description of the data analysis process, including measures to address common method bias (CMB) and validate the model, is commendable.

Suggestions for Improvement:

Clarify the rationale for selecting the specific constructs and indicators used for human capital, decent work, and integration. Were these based on prior validated scales, or were they developed specifically for this study?

Provide more details on the demographic characteristics of the sample, particularly regarding gender, age, and education levels. While Figure 2 provides an overview, a more detailed breakdown would enhance the reader's understanding of the sample's representativeness.

Discuss the potential limitations of using secondary data, particularly in terms of its applicability to the current research questions.

3. Results:

Strengths: The findings are clearly presented and supported by robust statistical analyses. The use of bootstrapping to validate the mediation analysis adds credibility to the results.

Suggestions for Improvement:

Expand on the implications of the low mediation effect of decent work. While the study acknowledges Peru's high labor informality, it would be helpful to explore additional factors that might explain this limited effect.

Include a discussion on the potential differences in integration outcomes for various subgroups within the sample (e.g., by gender, age, or length of residency).

4. Discussion and Policy Implications:

Strengths: The discussion effectively links the findings to broader policy implications, emphasizing the need for labor formalization and recognition of migrant skills.

Suggestions for Improvement:

Provide specific examples of policies or programs from other countries that have successfully addressed similar challenges. This would strengthen the practical relevance of the recommendations.

Discuss potential barriers to implementing the proposed policies in Peru and suggest strategies to overcome these challenges.

5. Ethical Considerations:

Strengths: The manuscript appropriately addresses ethical issues, including the use of secondary data from a publicly available database and approval by an ethics committee.

Suggestions for Improvement: Reiterate the steps taken to ensure the anonymity and confidentiality of participants, especially given the sensitive nature of migration data.

6. Writing and Presentation:

Strengths: The manuscript is well-written and follows a logical structure, making it easy to follow the arguments and findings.

Suggestions for Improvement:

Simplify technical terms and provide additional explanations for readers unfamiliar with PLS-SEM.

Ensure consistency in the use of terms like "human capital," "decent work," and "integration." A glossary or definition section could be helpful.

Additional Comments on Dual Publication, Research Ethics, or Publication Ethics:

Dual Publication: Ensure that no portion of this manuscript has been previously published or is under consideration elsewhere. While secondary data usage is appropriate, confirm that the authors have not presented identical analyses in other publications.

Research Ethics: The study relies on secondary data from a publicly available source. While the manuscript notes that consent was obtained, the authors should clarify whether additional permissions were required to reuse this dataset for the current study.

Publication Ethics: The manuscript appears to meet the ethical standards required for publication. However, the authors should confirm that all co-authors have approved the final manuscript and disclose any potential conflicts of interest.

Recommendation:

Based on the above evaluation, I recommend:

Minor Revisions to address the suggestions outlined above. The manuscript has significant potential, and these revisions will enhance its clarity, rigor, and relevance.

Summary of Suggestions:

Provide a more detailed comparison with similar studies in other regions.

Clarify the rationale for construct selection and address potential limitations of secondary data.

Expand on subgroup analysis and implications of the mediation effect.

Include examples of successful policies from other countries and discuss barriers to implementation in Peru.

Simplify technical terms and ensure consistency in terminology.

Confirm compliance with ethical standards and provide additional details on data reuse.

The findings of this study have the potential to contribute meaningfully to the field of migration and labor economics, particularly in the context of Latin America. Addressing the suggestions outlined above will strengthen the manuscript and enhance its impact.

6. PLOS authors have the option to publish the peer review history of their article (what does this mean? ). If published, this will include your full peer review and any attached files.

**Do you want your identity to be public for this peer review?** For information about this choice, including consent withdrawal, please see our Privacy Policy .

Reviewer #1: No

Reviewer #2: **Yes: ** DEJENDRAN RAJENRAN

---

## [Author Response · Author response to Decision Letter 1]

13 Mar 2025

Reviewers' comments:

Reviewer's Responses to Questions

Comments to the Author

“COMMENT”

1. Is the manuscript technically sound, and do the data support the conclusions?

Reviewer #1: Yes

Reviewer #2: Yes

“AUTHORS’ RESPONSE:”

We would like to thank both reviewers for recognizing it.

“COMMENT”

2. Has the statistical analysis been performed appropriately and rigorously?

Reviewer #1: Yes

Reviewer #2: Yes

“AUTHORS’ RESPONSE:”

We would like to thank both reviewers for recognizing it.

“COMMENT”

3. Have the authors made all data underlying the findings in their manuscript fully available?

The PLOS Data policy requires authors to make all data underlying the findings described in their manuscript fully available without restriction, with rare exception (please refer to the Data Availability Statement in the manuscript PDF file). The data should be provided as part of the manuscript or its supporting information, or deposited to a public repository. For example, in addition to summary statistics, the data points behind means, medians and variance measures should be available. If there are restrictions on publicly sharing data— e.g. participant privacy or use of data from a third party—those must be specified.

Reviewer #1: Yes

Reviewer #2: Yes

“AUTHORS’ RESPONSE:”

We would like to thank both reviewers for recognizing it.

“COMMENT”

4. Is the manuscript presented in an intelligible fashion and written in standard English?

Reviewer #1: Yes

Reviewer #2: Yes

“AUTHORS’ RESPONSE:”

We would like to thank both reviewers for recognizing it.

5. Review Comments to the Author

Reviewer #1: The article is prepared at a fairly high level. The research topic is important and relevant.

“COMMENT”

It is worth adding an assessment of the representativeness of the sample of respondents.

“AUTHORS’ RESPONSE:”

We´ve added the following information to the file:

“The data were taken from Cequea et al. (2024), so they were secondary data. The authors noted, with respect to the sample size “it was considered that 889,809 Venezuelans resided in Lima …, with a margin of error of 3 % and a confidence level of 95 %, having an estimated sample size of 1067 subjects. However, this study received a total of 1193 valid responses to the survey and 98 invalid ones.”(p. 5). However, the number of valid surveys exceeded the sample size, so it is considered that this sample size overcomes the drawbacks of non-probability sampling used for data collection in this study.”

“COMMENT”

It is worth adding the possibility of using the results in a business environment.

“AUTHORS’ RESPONSE:”

Thank you for your recommendation. We´ve added it to the new version.

“COMMENT”

Reviewer #2: Review of Manuscript PONE-D-24-51928

Title: Understanding the Effect of Human Capital and Decent Work for Migrants' Integration Using PLS-SEM

General Comments:

This manuscript addresses a critical and timely issue: the integration of Venezuelan migrants in Peru, focusing on the roles of human capital and decent work. The study utilizes Partial Least Squares Structural Equation Modeling (PLS-SEM) to explore the relationships among these variables, providing insights relevant for policymakers and stakeholders in migration and labor markets. The manuscript is well-structured, and its findings are significant for understanding migrant integration in contexts characterized by labor informality. However, there are several areas where the manuscript could be improved to enhance its clarity, rigor, and impact.

“AUTHORS’ RESPONSE:”

Dear Reviewer, we would like to extend our sincere gratitude for dedicating your time and expertise to evaluate our manuscript. Your insightful comments and constructive feedback have been instrumental in enhancing the quality and rigor of our work.

We have carefully considered each of the points raised, and in response, we have made the necessary revisions to address the concerns raised. We are confident that these revisions align with your expectations and have effectively clarified pertinent aspects of the manuscript.

We greatly value your contributions in ensuring the scholarly rigor and integrity of our research. We believe that the revisions made have strengthened the manuscript and we look forward to your further guidance as we move towards the publication stage.

Thank you once again for your invaluable input and for the opportunity to contribute to the body of knowledge in our field.

Specific Comments:

“COMMENT”

1. Research Context and Contribution:

Strengths: The study's focus on Venezuelan migrants in Peru addresses a gap in the migration literature, particularly in the context of Latin America's high labor informality. The emphasis on decent work as a mediating factor is innovative and aligns well with the United Nations’ Sustainable Development Goals.

Suggestions for Improvement: While the manuscript discusses the relevance of the study, it could provide a more detailed comparison with similar studies in other regions to highlight its unique contribution further. For example, are there lessons from other countries facing similar migration challenges that could enrich the discussion?

“AUTHORS’ RESPONSE:”

Thank you for the recommendation. We´ve added the following statement to the text:

“The literature review showed that there are numerous studies that relate human capital or the high skills of migrants with obtaining fair working conditions or decent work and this in turn as a facilitator of integration into the host society (); however, no studies were found that propose a direct relationship between human capital and integration considering decent work as a mediating variable, nor studies that employ structural equations or causal relationships between these constructs.

Peru's neighboring countries face similar migration challenges; but, although Venezuelans migrate fleeing violence, seeking work and improvements in their quality of life, the region is known for its high rates of informal employment, being this a structural problem, the incidence of this labor informality among Venezuelans abroad is particularly high. In Peru, 97% of work is in the informal sector, while in Panama, Ecuador, Colombia and Trinidad and Tobago, the Venezuelan labor force faces between 73% and 89% (R4V, 2025). Informality usually implies long working hours, lower wages, absence of social protection and related benefits, and often unsafe and precarious working conditions. In this context, various measures have been implemented in neighboring countries where Venezuelan migration has overwhelmed the capacity of local governance. To address this phenomenon, the Interagency Coordination Platform for Refugees and Migrants (R4V) was created to coordinate efforts under the Refugee and Migrant Response Plan for Venezuela (RMRP) in 17 countries in Latin America and the Caribbean. Countries such as Colombia, Ecuador and Chile, the main destinations of Venezuelan migrants and Peru's neighbors, have implemented various migration control strategies. These strategies include migration control policies, focused on regulating border flows; regularization, aimed at granting legal status to migrants; and security, focused on measures related to national and border security (R4V, 2025).”

2. Methodology:

Strengths: The use of PLS-SEM is appropriate for the study's objectives and the complexity of the model. The detailed description of the data analysis process, including measures to address common method bias (CMB) and validate the model, is commendable.

“COMMENT”

Suggestions for Improvement:

Clarify the rationale for selecting the specific constructs and indicators used for human capital, decent work, and integration. Were these based on prior validated scales, or were they developed specifically for this study?

“AUTHORS’ RESPONSE:”

Thank you for the recommendation. We´ve added the following statement to the text:

“The constructs and dimensions used to assess human capital, decent work and integration were based on previously validated scales and previously established theoretical frameworks (Cequea et al., 2024b).”

“COMMENT”

Provide more details on the demographic characteristics of the sample, particularly regarding gender, age, and education levels. While Figure 2 provides an overview, a more detailed breakdown would enhance the reader's understanding of the sample's representativeness.

“AUTHORS’ RESPONSE:”

This section was rewritten to provide more detail on the characteristics of the sample, as suggested. It may be noted at the text, as follows:

“Figure 2 provides a detailed overview of the sample's demographic characteristics. Women represent the majority of the sample (62%), while men comprise 38%. Regarding age distribution, the largest group falls within the 31–40 age range (37% for both genders), followed by participants aged 21–30 (28% of women and 34% of men). A smaller proportion of participants are above 50, with only 6% of men and 10% of women in the 51–60 age group, and even fewer aged 61 or older.

Marital status indicates that single individuals dominate the sample, accounting for 70% of women and 66% of men. Married participants represent 18% of women and 22% of men, while divorced and widowed individuals make up a small minority.

In terms of education, the sample reflects high human capital. Among women, 41% hold university degrees, compared to 34% of men. Postgraduate qualifications are less common, with 6% of women and 3% of men attaining this level. A notable proportion of men have secondary (40%) or technical education (22%), compared to 32% and 21% of women, respectively.

Employment patterns reveal a significant prevalence of independent work, particularly among women (46%) compared to men (36%). Meanwhile, 37% of women and 43% of men are employed without formal contracts, underscoring the informal nature of employment within the sample. A smaller subset of participants is employed under contractual agreements, with slightly more men (22%) than women (18%).

Residency duration in Peru varies across the sample. Most participants have lived in the country for three to five years, with 32% of women and 29% of men reporting 3–4 years of residence, and 28% of women and 39% of men indicating 4–5 years. These patterns suggest a mix of recently arrived individuals and those who are more established, which impacts labor market integration and stability.

Additional data highlight that 65% of respondents have 1–10 years of professional experience, indicating a relatively experienced workforce. Work intensity is notable, with 59% of participants working six days per week and 38% reporting shifts of 10–12 hours daily. This reflects the demanding nature of their work environments and economic necessity.

Lastly, 51% of respondents report practicing their profession in Peru, demonstrating a level of professional integration. However, this also implies that nearly half face barriers in exercising their qualifications, likely due to issues related to credential recognition or occupational mismatch.”

“COMMENT”

Discuss the potential limitations of using secondary data, particularly in terms of its applicability to the current research questions.

“AUTHORS’ RESPONSE:”

This discussion was incorporated into the item “6.2. Limitations and future research directions”, as follows:

“A limitation of secondary research is that the data were collected for another purpose and may not fully meet the needs of the current research. The objectives and methodology used to collect the primary data may not be appropriate for the problem at hand, so the quality of the data source must be carefully reviewed (Schiffman & Kanuk, 2005; Heinemann, 2007). For example, in the primary data collection (Cequea et al., 2024a), the data collection technique used did not guarantee parity between men and women, and the sampling was not random, so it is not possible to ensure representativeness. This could make it difficult to reach more robust conclusions if the objective of our study were aimed at making comparisons between groups of men and women; however, this was not the objective of the present study. On the other hand, to avoid assuming different meanings of the variables used for this secondary study, we reviewed the conceptualization and operationalization of these variables in the primary study (Cequea et al., 2024b) and adopted them for the secondary study.”

3. Results:

“COMMENT”

Strengths: The findings are clearly presented and supported by robust statistical analyses. The use of bootstrapping to validate the mediation analysis adds credibility to the results.

“AUTHORS’ RESPONSE:”

We would like to thank both reviewers for recognizing it.

“COMMENT”

Suggestions for Improvement:

Expand on the implications of the low mediation effect of decent work. While the study acknowledges Peru's high labor informality, it would be helpful to explore additional factors that might explain this limited effect.

“AUTHORS’ RESPONSE:”

Thank you for the recommendation. We´ve added the following statement to the text:

“According to the International Labor Organization (ILO, 1999), decent work implies access to productive opportunities in conditions of freedom, equity, security and human dignity, essential aspects for the social and economic integration of migrants in host societies, since it not only guarantees stable income and decent working conditions, but also promotes their social recognition, civic participation and general well-being. Likewise, access to employment with fair conditions allows migrants to develop economic autonomy, facilitating their stability and access to housing, education and health services. Additionally, it encourages the use of skills and knowledge acquired in the country of origin, avoiding overqualification and waste of talent, which reinforces their self-esteem and sense of belonging (Portes & Rumbaut, 2014). Finally, the workplace is a space where migrants can establish contact networks, learn cultural norms and develop interpersonal relationships that favor their social integration (Ager & Strang, 2008). In this sense, inclusive labor policies strengthen social cohesion and economic stability, benefiting both migrants and society as a whole, since when host countries guarantee fair labor conditions, they not only protect migrants' rights, but also take advantage of their human capital and promote more cohesive and equitable societies (Dustmann & Glitz, 2011).

Another aspect that could be affecting the low mediation of decent work are the barriers faced by migrants in Peru to regularize their qualifications and validate their knowledge and experience, finding themselves limited to underemployment and not being able to access well-paid jobs commensurate with their skills, which reduces their quality of life.”

“COMMENT”

Include a discussion on the potential differences in integration outcomes for various subgroups within the sample (e.g., by gender, age, or length of residency).

“AUTHORS’ RESPONSE:”

First of all, we would to thank you for the valuable recommendation. At this point, we would like to inform you that we were unable to respond to your suggestion due to not having access to the PLS software.

4. Discussion and Policy Implications:

Strengths: The discussion effectively links the findings to broader policy implications, emphasizing the need for

---

## [Editor Report · Decision Letter 1]

19 Mar 2025

Understanding the effect of human capital and decent work for migrants’ integration using PLS-SEM

PONE-D-24-51928R1

Dear Dr. Valentina Gomes Haensel Schmitt,

We’re pleased to inform you that your manuscript has been judged scientifically suitable for publication and will be formally accepted for publication once it meets all outstanding technical requirements.

Kind regards,

Voxi Heinrich Amavilah, Ph.D.

Academic Editor

PLOS ONE
---

## [Editor Report · Acceptance letter]

PONE-D-24-51928R1

PLOS ONE

Dear Dr. Schmitt,

I'm pleased to inform you that your manuscript has been deemed suitable for publication in PLOS ONE. Congratulations! Your manuscript is now being handed over to our production team.

Kind regards,

on behalf of

Dr. Voxi Heinrich Amavilah

Academic Editor

PLOS ONE